# Sex-Specific Differences in Toxicity Following Systemic Paclitaxel Treatment and Localized Cardiac Radiotherapy

**DOI:** 10.3390/cancers13163973

**Published:** 2021-08-06

**Authors:** Nicole Chmielewski-Stivers, Benoit Petit, Jonathan Ollivier, Virginie Monceau, Pelagia Tsoutsou, Ana Quintela Pousa, Xiaomeng Lin, Charles Limoli, Marie-Catherine Vozenin

**Affiliations:** 1Department of Radiation Oncology, University of California at Irvine, Irvine, CA 92697, USA; nchmiele@uci.edu (N.C.-S.); xiaoml3@uci.edu (X.L.); 2Laboratory of Radiation Oncology, Radiation Oncology Service, Department of Oncology, CHUV, Lausanne University Hospital, University of Lausanne, 1011 Lausanne, Switzerland; benoit.petit@chuv.ch (B.P.); jonathan.ollivier@chuv.ch (J.O.); Pelagia.Tsoutsou@hcuge.ch (P.T.); quintelapousa@gmail.com (A.Q.P.); 3Institut de Radioprotection et de Sureté Nucléaire (IRSN), 92260 Fontenay aux Roses, France; virginie.monceau@irsn.fr; 4Department of Radiation Oncology, Hôpitaux Universitaires Genèvehug (HUG), 1205 Geneva, Switzerland

**Keywords:** cancer treatment, chemotherapy, radiotherapy, cardiotoxicity, neurotoxicity, sex, *rhoB*

## Abstract

**Simple Summary:**

The objective of the present study was to investigate the impact of sex in the development of long-term toxicities affecting quality of life in cancer survivors after systemic paclitaxel treatment and cardiac irradiation. Sex-specific differences may affect tumor biology, drug pharmacokinetics and dynamics, and response to local treatment such as radiation therapy (RT). However, sex is rarely taken into consideration when administering cancer therapies. Interestingly, female mice are protected from paclitaxel-induced neurotoxicity as well as from radiotherapy-induced cardiotoxicity, and deficiency in the small GTPase *RhoB* reversed the protection in females but not in males. In conclusion, our results are the first to identify sex- and organ-specific responses to systemic paclitaxel administration and localized RT. These results may have important implications for the management of cancer patients and implementation of personalized medicine in oncology.

**Abstract:**

The impact of sex in the development of long-term toxicities affecting the quality of life of cancer survivors has not been investigated experimentally. To address this issue, a series of neurologic and cardiologic endpoints were used to investigate sex-based differences triggered by paclitaxel treatment and radiotherapy exposure. Male and female wild-type (WT) mice were treated with paclitaxel (150 and 300 mg/kg) administered weekly over 6 weeks or exposed to 19 Gy cardiac irradiation. Cohorts were analyzed for behavioral and neurobiologic endpoints to assess systemic toxicity of paclitaxel or cardiovascular endpoints to assess radiotherapy toxicity. Interestingly, female WT mice exhibited enhanced tolerance compared to male WT mice regardless of the treatment regimen. To provide insight into the possible sex-specific protective mechanisms, *rhoB-*deficient animals and elderly mice (22 months) were used with a focus on the possible contribution of sex hormones, including estrogen. In females, *RhoB* deficiency and advanced age had no impact on neurocognitive impairment induced by paclitaxel but enhanced cardiac sensitivity to radiotherapy. Conversely, *rhoB-*deficiency protected males from radiation toxicity. In sum, *RhoB* was identified as a molecular determinant driving estrogen-dependent cardioprotection in female mice, whereas neuroprotection was not sex hormone dependent. To our knowledge, this study revealed for the first time sex- and organ-specific responses to paclitaxel and radiotherapy.

## 1. Introduction

Epidemiological studies clearly show that sex has a significant impact on the biological phenotypes, disease progression, and treatment efficacy of cancer in oncology patients [1,2,3,4,5,6,7]. Surprisingly, very little basic research has sought to distinguish sex-differences in anti–cancer therapies, despite the obvious relevance such investigations would have on the optimization of treatment outcomes and the development of precision medicine in oncology. In addition, as advances in anticancer treatments continue to increase cancer survivorship [8], there is a growing concern about the long-term toxicities affecting the quality of life of patients surviving cancer. To date, no systematic investigation has evaluated the impact of sex on the short- and long-term complications induced by anticancer therapies, particularly at the normal tissue level. The present study was initiated to investigate these issues for two well-established anticancer strategies often used in combination for the treatment of breast cancer: paclitaxel and radiotherapy (RT). While this combination is used primarily to treat breast cancer, they are also used for the treatment of ovarian, pancreatic, head and neck and esophageal, and lung tumors and display distinct and well-reported toxicity profiles due to their fundamentally distinct modes of action [9,10]. However, as alluded to above, a critical side-by-side comparison of such toxicities between the sexes has yet to be reported.

Paclitaxel is among the most profitable and widely used chemotherapeutics in history [11]. Originally discovered and isolated from the Pacific yew tree, paclitaxel is part of the taxane family of chemotherapeutics that works as a microtubule poison, promoting microtubule assembly and preventing depolymerization through tubulin subunit binding [12,13,14]. Despite this extensive research history and expanding use of paclitaxel in the treatment of cancer [11], many uncertainties remain concerning the fundamental mechanisms of cancer-specific paclitaxel toxicities, while even less is known about its effects on normal tissue [9,15]. Regarding normal tissue toxicity, the occurrence of neurotoxicities as well as neuropathies is well described in breast cancer patients undergoing paclitaxel treatment [16,17]. Pneumopathies are also reported [18], as well as cardiac complications, including sinus bradycardia, conduction blocks, ventricular tachycardia, and ischemic manifestations [19,20]. In the CNS, the highly lipophilic nature of paclitaxel enables its crossing of the blood–brain barrier. Despite rapid clearance from the CNS by an active p-glycoprotein-mediated transporter mechanism, paclitaxel treatment still induces significant short- and long-term postchemotherapy cognitive impairment in breast cancer patients [21]; however, to date, few studies have investigated the impact of sex on neurocognitive function after paclitaxel treatment.

The toxicity of radiotherapy is localized to healthy tissues surrounding tumors that are irradiated at the time of anticancer treatment. In the context of left-sided breast cancer, the main organ at risk is the heart, and cardiotoxicity has been reported in epidemiological studies performed in large cohorts of breast cancer patients [10]. Physiopathological investigations [22] are also available and provide insights about the pathogenic imprint associated with radiation-induced cardiac toxicity. Surprisingly, although in the cardiovascular field sex-hormone-mediated cardioprotection of female patients is well known [23,24,25,26], no investigation related to the impact of sex is available after exposure to RT.

Therefore, the aim of the present study was to investigate the impact of sex following either systemic paclitaxel treatment or cardiac irradiation. First, the impact of paclitaxel treatment on the general health status and survival of male and female wild-type (WT) mice was investigated along with more specific effects on neurocognitive function. Second, the impact of RT on the general health status and survival of male and female WT mice was investigated along with more specific effects on cardiac function. In each case, females were more tolerant than males to either treatment. Next, we used a genetic model (*rhoB*^–/–^) and aged female mice (22 months) to probe how reduced hormone levels and signaling altered the response of female mice to cancer treatments. Interestingly, we found cardioprotection after RT, but not neuroprotection after paclitaxel, effects that were oppositely regulated in female versus male mice by the *rhoB* pathway, pointing to the regulatory role of the ERα signaling cascade in female and Connective Tissue Growth Factor (CTGF) in male mice.

## 2. Materials and Methods

### 2.1. WT and rhoB^–/–^ Mice

All animal procedures and necropsy were approved by the institutional animal care and use committees of University of California Irvine (UCI) and Lausanne University hospital, Lausanne, CH (VD3189). Wild-Type (WT) animals were either purchased from Jackson Labs (Sacramento, CA, USA) or provided by on-campus breeding facilities (UNIL/CHUV). All animals were housed with ad libitum standard diet and water on a 12-h light cycle. All animal-related handling and experiments were performed during standard light hours. Animal weight was measured every week. Vaginal cytology was performed on all the female animals prior and during behavior weeks to confirm reproductive status.

### 2.2. Paclitaxel Treatment

Adult male and female mice (C56Bl6/J and *rhoB*^–/–^), ranging from 12 to 88 weeks of age, were divided into vehicle- (0.02% ethanol made in sterile saline) and paclitaxel-treated groups. Paclitaxel was administered IP once a week over 6 weeks at a dose of up to 150 mg/kg or 300 mg/kg (paclitaxel, Carbosynth, Compton, UK).

### 2.3. Irradiation Procedures

Heart irradiation was performed with a small animal imaged-guided irradiator, the X-RAD 225 Cx (Pxi Precision X-ray, North Branford, CT, USA). The isocenter was placed in the heart using cone beam computed tomography, (Precision X-Ray, North Branford, CT, USA). Irradiation was performed with 2 beams parallel-opposed a 15 mm circular collimator (Precision X-Ray, North Branford, CT, USA) at 225 Kev, 13 mA, with a 0.3 mm copper filter. Treatment was delivered using a treatment planning system (SmART-Plan, SmART Scientific Solution, Maastricht, The Netherlands). For irradiation, anesthetized mice were immobilized in a feet-first supine position.

### 2.4. Behavior Testing

All tests were performed in designated behavior rooms with ambient noise during light hours. Fear extinction was recorded and analyzed using the Habitest Modular System (Harvard Apparatus, Holliston, MA, USA) and FreezeFrame (Coulbourn Systems, Holliston, MA, USA) software. All tasks were performed and recorded by the same individual as described previously [27]. Open-Field Test (OFT) was performed in a fully lit behavior room, mice were placed in an empty white arena with no habituation or prior extensive handling. Exploration and distance traveled were measured over 5 min using Ethovision XT (Noldus, Leesburg, VA, USA) software as described previously [28]. Light Dark Box (LDB) was performed in a fully lit behavior room: mice were placed in an arena with an open white ‘light’ portion connected to a closed black ‘dark’ compartment. Transitions between each compartment were recorded over 10 min using Ethovision XT (Noldus) software as described previously [27].

Fear Extinction (FX) memory test [27] was performed in a dark room lit only by a red light. Mice were placed in a closed chamber and received 3 tone-associated electric shocks. Twenty-four hours later they were placed in a different chamber context, in the same room, for 3 consecutive days where the tone was played with no shock. After completion of 3 days of extinction training, for the final ‘Extinction Test’ day, mice were placed back to the original tone-shock context and were presented with three tones at two minute intervals without receiving shock. Percent freezing was recorded to assess animals’ ability to extinguish original fear memory.

### 2.5. Age and Behavior Testing Timeline

Animals underwent behavior testing after paclitaxel exposures. Age of animals, week of behavioral testing during study, and N animals tested are specified in Table 1.

### 2.6. Transthoracic Ultrasound Imaging (Echocardiography)

Transthoracic ultrasound imaging was performed using the MS400 (18–38 MHz) probe from Vevo 2100 color Doppler ultrasound machine (VisualSonics, Toronto, ON, Canada). Mice are lightly anesthetized with 1–1.5% isoflurane (Provet, Lyssach, Switzerland), maintaining heart rate at 500–600 beats per minute. The mice were placed in decubitus dorsal on a heated 37 °C platform to maintain body temperature. A topical depilatory agent was used to remove the hair, and ultrasound gel was used as a coupling medium between the transducer and the skin. The heart was imaged in the 2D mode in the parasternal long-axis view. From this view, an M-mode cursor was positioned perpendicular to the interventricular septum and the posterior wall of the left ventricle at the level of the papillary muscles. The measurements were taken in three separate M-mode images and averaged. Left ventricular Ejection Fraction (%EF) was also calculated. %EF is derived from the formula of EF (%) = [(LVDA LVSA)/LVDA] × 100.

### 2.7. Histology

Cardiac tissue was paraffin embedded and transversely cut using a scientific microtome (Thermo Fisher Scientific, Illkirch, France) into 5 µm sections and underwent ethanol dehydration and H&E tissue staining.

### 2.8. Western Blot Analysis

Frozen cardiac tissue was processed using a tissue homogenizer in RIPA Lysis Buffer System^®^ (Life Technologie, Zug, Switzerland). Quick Start™ Bradford Protein Assay (Bio-Rad Laboratories, Cressier, France) was used for protein lysate concentration measurements. Rabbit anti-estrogen receptor α primary antibodies (1:1000, ab75635, Abcam, Cambridge, UK) and mouse anti-GAPDH (1:40,000, mab374, Merck, Schaffhausen, Switzerland) were measured. Membranes were imaged using FusionFX (Vilber, Marne-la-Vallée, France).

### 2.9. Data Analysis

Statistical analyses were conducted using GraphPad Prism (v8.4, GraphPad Software, San Diego, CA, USA). Unpaired, two-tailed tests were used to compare behavioral results with two groups. Two-way analysis of variance (ANOVA) was used to assess for interaction and group effects of more than two groups for behavioral and echocardiography testing, if significance was found, Bonferroni’s multiple comparisons test was conducted. Three-way ANOVA was used to compare four groups, two-way ANOVA was used to compare two groups of weight data, and no multiple comparisons assessment was performed on weight data. Log-rank (Mantel–Cox) test was used to compare survival curves. *p* ≤ 0.05 was considered statistically significant. 

## 3. Results

### 3.1. Female-Specific Protection against Neurotoxic Effects of Paclitaxel

We investigated systemic and neurotoxicity induced by paclitaxel injection in adult male and female Wild-Type C57Bl6 (WT) mice. Interestingly, data indicated a female-specific protection against paclitaxel-induced toxicity in terms of survival, weight, and behavioral effects.

In male mice (*n* = 18), the lower dose (150 mg/kg) of paclitaxel resulted in a 25% reduction in survival, whereas in females (*n* = 18), lethality was not observed (Figure 1Ai). Enhanced toxicity of paclitaxel was also evident in males that lost significantly more weight compared to females over the course of the 15-week study (three-way ANOVA, significant effect on sex (*p* < 0.0001), treatment (*p* < 0.0001), time × treatment (*p* < 0.0001), sex × treatment (*p* < 0.0001), and time × sex × treatment (*p* < 0.0001) (Figure 1Aii). Behavioral testing showed a greater sensitivity in WT males compared to females. Acquisition of conditioned fear was not impaired in treated or control female cohorts that demonstrated a gradual decrease in freezing behavior over extinction sessions (Figure 1Aiii). Similarly, paclitaxel treatment (150 mg/kg) did not impair fear acquisition in males, but compared to controls, treated males did not exhibit the gradual decrease in freezing behavior over extinction sessions (Figure 1Aiv). Importantly, a fear extinction test conducted 72 h after training demonstrated a significant impairment in extinction memory in males, but not in females (two-way ANOVA interaction effect *p* = 0.0026 and treatment effect *p* = 0.046), as well as significantly more freezing in paclitaxel-treated males compared to paclitaxel-treated females and control males (multiple comparisons and Bonferroni test effects for female vs. male treated, *p* = 0.047 and male control vs. treated, *p* = 0.0041) (Figure 1Av). There was no effect of paclitaxel treatment on distance traveled in males or females during open-field testing, suggesting paclitaxel had no significant effect on locomotor activity of animals (Figure 1Avi). However, paclitaxel-treated males transitioned fewer times between the compartments during light dark box testing, suggesting increased anxiety-like behavior after 150 mg/kg paclitaxel treatment in male, but not female, WT animals (two-way ANOVA interaction effect *p* = 0.029 and sex effect *p* = 0.0089; multiple comparisons Bonferroni test effects for female vs. male treated *p* = 0.0049 and male control vs. treated *p* = 0.0122) (Figure 1Avii).

The higher dose of paclitaxel (300 mg/kg) achieved a lethal dose 100 (LD_100_) in males by week 10 of the study (log-rank Mantel–Cox test demonstrated significant differences in survival between treatment and sex, *p* < 0.0001) (Figure 1Bi). WT males demonstrated considerably more weight loss (13%) compared to females after paclitaxel treatment, necessitating an early termination of the study at week 10 (three-way ANOVA significant effect found for sex (*p* < 0.0001), treatment (*p* < 0.0001), time × treatment (*p* = 0.0043), sex × treatment (*p* < 0.0001), and time × sex × treatment (*p* < 0.0001)). (Figure 1Bii). At 300 mg/kg, behavioral deficits were demonstrated in both male and female mice. Acquisition of conditioned fear was not impaired in treated or control female cohorts that demonstrated a gradual decrease in freezing behavior over extinction sessions (Figure 1Biii). In males, 300 mg/kg treatment did not impair fear acquisition, but similarly to the 150 mg/kg treatment, males failed to gradually decrease freezing behavior over extinction sessions (Figure 1Biv). Both females and males treated with 300 mg/kg experienced compromised fear extinction during fear extinction tests (two-way ANOVA sex effect *p* < 0.0001 and treatment effect *p* < 0.0001), as well as significantly more freezing between control vs. treated females (*p* = 0.0041), control vs. treated males (*p* = 0.0041), control females vs. control males (*p* = 0.0002), and treated females vs. males (*p* = 0.0002) according to two-way ANOVA multiple comparisons, Bonferroni analysis (Figure 1Bv). Due to significant male-specific toxicity to paclitaxel at 300 mg/kg, experiments were terminated just following fear extinction, which precluded additional behavioral testing. As a result, males were then sampled 10 weeks after the start of treatment. For the female cohort, behavioral testing continued and demonstrated significant differences in distance traveled during open-field testing, suggesting that the higher paclitaxel dose may have impacted locomotor activity (unpaired *t*-test, *p* < 0.0001) (Figure 1Bvi). In addition, paclitaxel-treated WT female mice transitioned fewer times between the compartments during light dark box testing, suggesting a possible increase in anxiety-like behavior after 300 mg/kg paclitaxel treatment (unpaired *t*-test, *p* = 0.0054) (Figure 1Bvii).

### 3.2. Female-Specific Protection against Cardiotoxic Effects of Radiotherapy

To further characterize sex differences following anticancer therapies, a single high dose of 19 Gy local irradiation, known to elicit cardiotoxicity, was given to adult WT male and female mice. Long-term cardiotoxicity was investigated 53 weeks post-RT, where 75% of males compared to only 20% of females died (log-rank (Mantel–Cox) test *p* = 0.0021), showing a female-specific protection from cardiac irradiation (Figure 2A).

Echocardiography was used to monitor heart function in animals. In WT females, cardiac function monitored by the ejection fraction (%EF) was similar in control and irradiated animals (data not shown), whereas irradiated WT males exhibited a 25% and 15% decrease in EF at 20 and 50 weeks post-RT, respectively, as compared to age-matched controls (*p* = 0.003) (Figure 2B). Histopathological analysis confirmed the functional results obtained by ultrasound. No significant structural alteration of cardiac ventricles was found in females until 50 weeks post-RT (Figure 2C). On the contrary, a dramatic disruption of ventricular structure associated with cardiomyocyte atrophy was observed 20 weeks post-irradiation. This pathogenic pattern worsened 50 weeks post-RT. Large patches of scar tissue infiltrating the septum and right ventricle were observed, indicative of delayed and massive death of cardiomyocytes. Resultant replacement amyloïdosis/fibrosis altered proper contractility of the cardiac muscle and caused loss of cardiac function as measured by ultrasound (Figure 2C).

### 3.3. RhoB Deficiency and Age-Related Reduction in Sex Hormone Do Not Impact Neuroprotection in Females Treated with Paclitaxel

Given the known differences in hormonal status between males and females, it was logical to address the potential role of the estrogen pathway on neurological outcomes. To avoid the use of pharmacological compounds (i.e., anti-estrogen such as anti-aromatase) that could interfere with long-term outcome, a genetic approach was taken that utilizes *rhoB*-deficient animals as well as old female mice having naturally low levels of sex hormones. The small Rho GTPase *RhoB* is known to mediate the activation of estrogen receptor alpha, which activates various downstream signaling cascades [29] evolved to regulate cell proliferation, differentiation, metabolism, and survival. Therefore, by decreasing ERα levels and/or activation (Appendix A) to impair downstream signaling, we speculated that *rhoB-*deficient females would be protected from paclitaxel-induced toxicity while *rhoB* deficiency and old age would reverse female-specific protection by disrupting hormone signaling cascades.

In order to test this hypothesis, *rhoB*^–/–^ females were treated with 150 and 300 mg/kg of paclitaxel (25 mg/kg × 6 weeks, intra-peritoneal). At the lower dose of 150 mg/kg, no impact on female survival was observed, although decreases in weight gain were found (Figure 3A,B). At the higher dose of 300 mg/kg, delayed death of *rhoB*^–/–^ females was found as compared to WT females, although similar numbers of female WT and *rhoB*^–/–^ mice (75%) were alive at the end of the study (13 weeks post-treatment). Weight gain was not detected in WT and *rhoB*^–/–^ animals given the higher dose of 300mg/kg paclitaxel, three-way ANOVA demonstrated significant variation due to treatment (*p* < 0.0001) and genotype (*p* < 0.0001) after 300 mg/kg (Figure 3C,D). Further, high-dose paclitaxel treatment did not modify cognitive behavior in *rhoB-*deficient females compared to controls (Figure 3E). Fear acquisition of freezing behavior over extinction sessions was not impaired in *rhoB-*deficient treated females, and there was no decrease in freezing during extinction training in control or treated *rhoB-*deficient females (Figure 3Ei). The fear test demonstrated no difference in the capability of treatments to impact fear extinction memory (Figure 3Eii). There was also no difference in mobility (Figure 3Eiii) or anxiety behavior (Figure 3Eiv) after 300 mg/kg paclitaxel in *rhoB-*deficient female mice. These results suggest that *rhoB*-mediated impairments in ER signaling pathway are not involved in paclitaxel-induced neurotoxicity. Since WT males could not tolerate the higher dose of paclitaxel, *rhoB*^–/–^ males were only subjected to 150 mg/kg paclitaxel. Control experiments performed on male WT and male *rhoB-*deficient mice at 150 mg/kg paclitaxel showed similar numbers (85%) of animals surviving at the end of the study (15 weeks post-treatment) (Appendix A).

To further explore the contribution of female hormones on neurocognition after paclitaxel treatment, 9–11-month-old and 22-month-old female mice possessing reduced sex hormone levels were treated with 150 mg/kg paclitaxel. No impact of 150 mg/kg paclitaxel was observed in 9–11-month-old females, but 29% of the older cohort died 11 weeks after paclitaxel treatment, which was associated with a dramatic loss of weight, indicating an enhanced sensitivity of the 22-month-old females to paclitaxel (Figure 4A,B). Surprisingly, paclitaxel treatment did not impair cognitive behavior in these older female cohorts compared to controls (Figure 4C). Fear acquisition of freezing behavior over extinction sessions was not impaired in either of the older female cohorts, and there was a gradual decrease in freezing during extinction training in control or treated *rhoB-*deficient females (Figure 4Ci,ii). The fear test demonstrated no difference in the capability of paclitaxel treatment to impact fear extinction memory (Figure 4Ciii). There was also no difference in mobility (Figure 4Civ) or anxiety behavior (Figure 4Cv), again pointing to the lack of involvement of female sex hormones in paclitaxel-induced neurotoxicity.

### 3.4. RhoB Deficiency Triggers Opposite Effects in Female and Male Mice Exposed to Radiotherapy

In females, estrogen is known to be a major cardioprotective factor, which provides the rationale for exploring whether *rhoB* deficiency would decrease the female-specific protection against RT-induced cardiotoxicity observed in WT female mice. Indeed *rhoB*^–/–^ females exhibited an increased sensitivity to RT exposure with death occurring at early time points, reaching an LD_50_ at 16 weeks and an LD_100_ at 30 weeks, while 80% of WT females survived to the end of the study 50 weeks post-RT (Figure 5A).

While alterations in cardiac function were not identified by ultrasound (Figure 5B, unpaired *t*-test *p* = 0.0231), histological analyses did reveal enhanced infiltration of leukocytes in cardiac ventricles in *rhoB*^–/–^ females 20 weeks post-RT (Figure 5C). p65 subunit of NF-κB was identified as one possible cardioprotective mediator downstream the *RhoB* and ERα cascade. The level of p65 was found to be lower in *rhoB*-deficient vs. WT females after RT. Interestingly, in males, the reverse was observed, as *rhoB*^–/–^ males were more tolerant to RT whereas 75% of the WT males died post-RT (Figure 5D). Further, no alteration of cardiac function was found in *rhoB*^–/–^ males by ultrasound (data not shown), and no histologic damage or fibrosis infiltration was detected (Figure 5E). Importantly, these results identified for the first time a sex-specific difference in response to RT, with *rhoB* as a major molecular determinant. In females, *rhoB* might mediate cardioprotection after RT via activation of the estrogen pathway, whereas in males, *rhoB* might mediate cardiotoxicity via activation of the CTGF fibrogenic pathway [30].

## 4. Discussion

Recently, sex-related differences in cancer incidence have increasingly been recognized and attributed to regulation at the genetic and molecular level and to sex hormones such as estrogen. Surprisingly, and despite the emergence of tailored treatments and personalized medicine, the impact of sex on therapeutic outcome and tolerance to treatment to date has been poorly investigated. These investigations are of utmost relevance since sex-specific differences may not only affect tumor biology but also the pharmacokinetics and dynamics of drugs and response to local treatment such as radiotherapy. In this context, the present experimental study was designed to investigate the impact of sex following two classical anticancer treatments, i.e., paclitaxel and RT. Both are widely used and administered according to the type and grade of cancer, but importantly, not according to the sex of the patient. Taking thoracic malignancies as a paradigm, the impact of paclitaxel on neurocognition and of RT on cardiac function were investigated in two distinct cohorts of female and male mice. Interestingly, our results are the first to show that female mice are protected from paclitaxel-induced neurotoxicity as well as from radiotherapy-induced cardiotoxicity. We also identified *rhoB* as an organ and sex-specific molecular determinant. Neuroprotection was not *rhoB-*mediated in either sex, whereas cardioprotection was *rhoB-*mediated in an opposite manner between female and male mice.

Accumulating evidence supports sex-related response to chemotherapeutic agents with differences in efficacy and toxicity (review [6]). For instance, Joerger et al. reported significant variation in the pharmacodynamics of paclitaxel in female patients with solid tumors compared to male patients. Females exhibited 20% lower elimination of paclitaxel than male patients did, with their peripheral compartments saturated at lower plasma concentrations (0.83 female vs. 1.74 mmol/L male) and paclitaxel elimination was slower (1 h female vs. 0.5 min male) [31]. Enhanced sensitivity of female patients to paclitaxel is further supported by the high number of female patients exhibiting severe leukopenia upon combined treatment with paclitaxel and carboplatin [5,32]. Conversely, female patients diagnosed with lung carcinoma who were treated with paclitaxel combined with carboplatin showed longer median progression-free survival (PFS) rate than male patients did [5,32].

Our present investigation focused on the adverse neurocognitive consequences of paclitaxel as neurotoxicity is known to be one of the more prominent side effects of paclitaxel that can have a dramatic impact on a patient’s quality of life. This question was investigated using a series of cognitive tests to interrogate locomotor activity, anxiety-like behavior, and fear extinction as a higher-order measure of cognitive flexibility. The five-day context-dependent fear extinction task used examined the ability to acquire and then extinguish a tone-associated fear memory. Whereas the ability to acquire a fear memory is predominantly amygdala mediated [33], long-term (four day) extinction recruits medial prefrontal cortex activity [34], engaging in translationally relevant executive function necessary for managing quality-of-life in cancer therapy recipients. 

Present data indicate that female mice were more tolerant than males to higher doses of paclitaxel, selected to be comparable with dosages used in breast cancer patients [35]. Our results are consistent with a recent study by Liang et al., who found that at a lower peripheral neuropathy-inducing dose of paclitaxel (16 mg/kg), female C57B6 mice exhibited protection against behavioral deficits compared to males but both sexes showed mechanical pain hypersensitivities, indicative of peripheral neuropathy [36]. Liang et al. also explored the molecular signature and found altered gene expression related to neurotransmission suggesting dysfunction in the medial prefrontal cortex (mPFC), known to be critical in both positive and negative regulation of extinction memory [37]. The other reported mechanisms of paclitaxel-induced neurotoxicity involved axonal/neuronal mitochondrial dysfunction, altered calcium homeostasis and calcium channel expressions, changes in peripheral nerve excitability, including altered expression and function of ion channels, immune dysfunction, and neuroinflammation at the level of axons, dorsal root ganglia, and within the spinal cord [38], as well as direct effects on brain tissues [39,40,41,42].

In the present study, we chose a more physiological and functional approach to investigate the contribution of female hormones. While the majority of published studies have focused on the contribution of testosterone [36], the role of female hormones has been less investigated. Notwithstanding, chemotherapeutic agents including paclitaxel are known to accelerate menopause and induce drops in hormone levels associated with cognitive impairments. The precise mechanisms linking estrogen to cognition are however complex, as estrogen supplementation has been shown to improve cognitive function [43,44]. Wang et al., for example, observed alternative downstream estrogen receptor-mediated signaling activity depending on endogenous or exogenously administered estrogen facilitating memory-dependent long-term potentiation in female rat brain tissue [45]. Based on the foregoing, we decided to implement a genetic approach through the use of *rhoB-*deficient mice that disrupts estrogen signaling and thought to use aged females with naturally lower levels of female hormones. Interestingly, data indicated that improved performance on select cognitive tasks in female mice was not dependent on *rhoB-*dependent estrogen signaling or female hormones. Using either model (*rhoB-*deficient or aged female mice), neurocognitive impairments observed after Paclitaxel treatment were not enhanced. While the exact mechanisms associated with this enhanced neurologic tolerance will require further investigation, immune regulation as suggested by Liang et al., or increased resistance of oligoprogenitor cells (OPC) responsible for maintaining the state of CNS myelination are plausible possibilities.

In addition to neurological side effects, cardiotoxicity of various cancer therapies defines another major concern in the field of oncology. Specifically, cardiac damage induced by RT remains a critical dose-limiting factor [46,47,48,49] despite recent advancements in treatment planning and image-guided radiation therapy. Additionally, as the number of long-term cancer survivors is increasing, complications emerge that can dramatically impair quality of life. Whereas over the past four decades, research has enhanced our understanding of the patho-physiological, cellular, and molecular processes governing radiation-induced cardiac toxicity [22], the impact of sex remains relatively undefined and under investigated [50] In clinical and preclinical studies, proper comparisons between female and male patients have never been done as they have not included corrections for the volume of irradiation [50], and only a few preclinical studies have ever included both sexes. Until now, experimental studies on radiation-induced cardiotoxicity were mainly conducted in male mice [51,52,53,54], largely based upon assumptions coming from the cardiovascular field stating that females should be more resistant than males to cardiac diseases. However, cardiovascular disease in women remains understudied, underdiagnosed, and undertreated, and even now, women remain under-represented in clinical trials related to cardiac disease [55]. For nearly 20 years, the contribution of estrogen has been identified in the pathogenesis of heart failure because postmenopausal women have increased risk of developing cardiac diseases [56]. Estrogen has been shown to attenuate the development of pressure overload-induced hypertrophy in mice [57,58] through regulation of nitric oxide synthase (NOS) [59].

In this context, and to parallel our studies with paclitaxel, the role of *rhoB* deficiency in cardiac toxicity was investigated in both sexes by image-guided focal radiotherapy. Present findings revealed an enhanced tolerance of C57Bl6 female mice to RT, as compared to the male mice. Data presented here also support the role of estrogen in female tolerance [23,24,25,26], as *rhoB-*deficient females having disrupted estrogen signaling through ERα are sensitized to radiotherapy. While limited sample sizes of aged female mice precluded more comprehensive cardiac investigations in the present study, our findings with *rhoB-*deficient male mice found them to be protected from radiation injury.

Interestingly, our study revealed opposite effects of *rhoB* deficiency on cardiac outcomes in female and male mice. Whereas in females, *rhoB* deficiency short-circuited the protective estrogen pathway, in males it disrupted the deleterious fibrogenic pathway dependent upon RhoB/CTGF identified in our previous study [51]. Consistent with these prior findings, no fibrosis deposition was observed and cardiac function was maintained. Whether this observation is specific to this strain of mouse will require further investigation, but our findings highlight the need to design future investigations to directly assess the impact of sex on organ-specific toxicities induced by anticancer therapeutic regimens.

Further work is clearly required to unravel the mechanistic basis of our findings and explore how they might impact personalized treatments in clinical practice. Neurocognitive assessments were undertaken to survey the impact of systemic paclitaxel across multiple brain regions, but the precise levels of paclitaxel crossing the blood–brain barrier into discrete regions and among the different age groups were not assessed. More importantly, the impact of combined treatment (paclitaxel + radiotherapy) was not investigated, whereas the combination is expected to enhance cardiac toxicity and possibly influence neurocognitive outcome by the release of paracrine factors. Future studies undertaking direct measurement of circulating and tissue-specific estrogen levels along with other nongenetic targeted interventions aimed at dissecting fibrogenic signaling cascades would also provide deeper mechanistic insight into tissue remodeling processes affecting the heart and surrounding organs (e.g., esophagus and lung).

## 5. Conclusions

In conclusion, our results are the first to identify sex- and organ-specific responses to systemic paclitaxel administration and localized RT with enhanced tolerance in WT females. These results may have important implications for the management of cancer patients and implementation of personalized medicine in oncology. Further investigations should include female and male cohorts and the influence of combined strategies that may encompass unforeseen off-target effects.

## Figures and Tables

**Figure 1 cancers-13-03973-f001:**
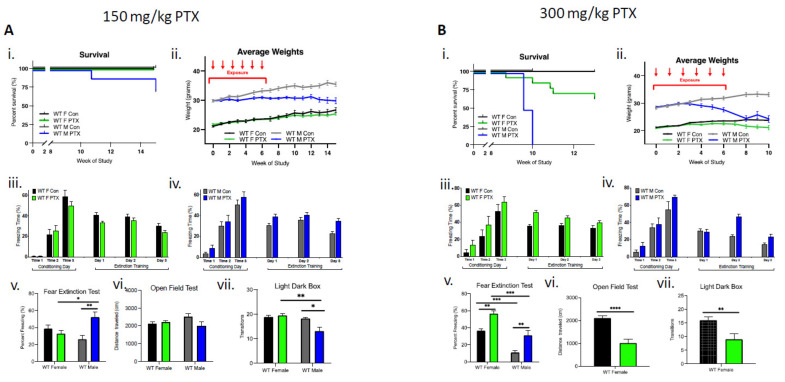
Wild-type females are more tolerant than males to paclitaxel and are protected from neurotoxicity induced by paclitaxel. (**A**) After 150 mg/kg paclitaxel treatment, (**i**) survival curves show 18% of WT males (*n* = 18) died while all females lived (*n* = 18). No significance in survival curve log-rank (Mantel–Cox) test. (**ii**) Females showed no systemic impact of paclitaxel treatment on the evolution of their weight, whereas males did not exhibit weight uptake. Significant variation in sex (*p* < 0.0001), treatment (*p* < 0.0001), time × treatment (*p* < 0.0001), and sex × treatment (*p* < 0.0001). (**iii**) Paclitaxel administration in WT female mice did not impair the acquisition of conditioned fear (three tone-shock pairings). All mice showed a graduate decrease in freezing behavior over extinction sessions (tone only) (*n* = 8). (**iv**) Paclitaxel administration in WT male mice did not impair the acquisition of conditioned fear (three tone-shock pairings). Controls, but not treated male mice showed a graduate decrease in freezing behavior over extinction sessions (tone only) (*n* = 8). (**v**) WT females, but not males, successfully abolished fear extinction memory (two-way ANOVA interaction effect *p* = 0.0026 and treatment effect *p* = 0.0457, *n* = 8). (**vi**) Paclitaxel treatment had no impact on spontaneous exploration (females: control *n* = 11, treated *n* = 18; males: control *n* = 12, treated *n* = 16). (**vii**) Treated males exhibited significantly more anxiety behavior (two-way ANOVA interaction effect *p* = 0.029 and sex effect *p* = 0.0089; females: control *n* = 7, treated *n* = 8; males: control *n* = 12, treated *n* = 16). (**B**) (**i**) After 300 mg/kg paclitaxel treatment, lethal dose 100 occured after 10 weeks in male mice (*n* = 12), whereas females (*n* = 14) exhibited 14% lethality after 10 weeks. Log-rank (Mantel–Cox) test comparison of survival curves significantly different between sexes (*p* < 0.0001). (**ii**) Females showed no weight uptake, males exhibited a drop of weight, with significant variation in sex (*p* < 0.0001), treatment (*p* < 0.0001), time × treatment (*p* < 0.0001), sex × treatment (*p* < 0.0001), and time × sex × treatment (*p* < 0.0001). (**iii**) Paclitaxel administration in WT female mice did not impair the acquisition of conditioned fear (three tone-shock pairings). All mice showed a graduate decrease in freezing behavior over extinction sessions (tone only) (*n* = 8). (**iv**) Paclitaxel administration in WT male mice did not impair the acquisition of conditioned fear (three tone-shock pairings). Controls, but not treated mice, showed a graduate decrease in freezing behavior over extinction sessions (tone only) (*n* = 6). (**v**) Paclitaxel treatment compromised fear extinction abolishment on test day (two-way ANOVA sex effect *p* < 0.0001 and treatment effect *p* < 0.0001; females *n* = 8, males *n* = 6). (**vi**) Paclitaxel treatment significantly decreased spontaneous exploration behavior in female WT mice (*p* < 0.0001; control *n* = 14, treated *n* = 12). (**vii**) Treated females exhibited significantly more anxiety behavior (*p* = 0.0054; control *n* = 14, treated *n* = 10). (**A,B**) Graphs designate mean ± s.e.m; *p-*value derived from unpaired *t*-test (**Bvi**–**vii**), three-way ANOVA (**Aii**,**Bii**), two-way ANOVA (**Aiii**–**vii**, and **Biii**–**v**), Bonferroni test (**Av**, **Avii**, and **Bv**): * *p* < 0.05, ** *p* < 0.01, *** *p* < 0.001, **** *p* < 0.0001.

**Figure 2 cancers-13-03973-f002:**
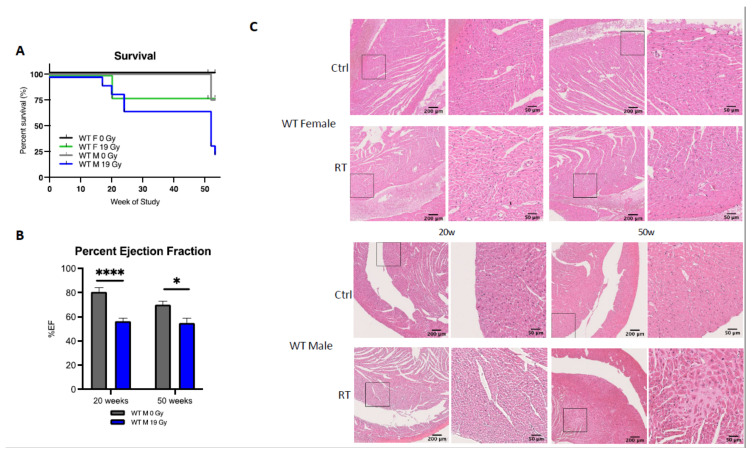
Wild-type females are protected from cardiotoxicity induced by RT (19 Gy). (**A**) Survival curves show that 77% of WT males died (*n* = 9), while 25% of WT females (*n* = 12) were alive 50 weeks post-19 Gy thoracic irradiation. Log-rank (Mantel–Cox) test *p* = 0.0021. (**B**) Significant change in %EF after exposure was shown in WT males (*p* = 0.0003). Multiple comparisons measure significant changes after 19 Gy at 20 weeks (*p* < 0.0001) and 50 weeks (*p* = 0.0127). Graphs designate mean ± s.e.m; *p-*value derived from multiple comparisons mixed-effect model (REML), Bonferroni test: * *p* < 0.05 and **** *p* < 0.0001. (**C**) Representative images of H&E-stained cardiac tissue showed no alteration of cardiac structure 20 weeks post-RT in females and males with a sex-specific evolution in time. Alterations of the cardiac structure were observed in males 50 weeks post-RT with larges patches of replacement amyloidosis/fibrosis deposition and scarring tissue associated with immune infiltration, whereas in females, cardiac structure remained quasi-normal.

**Figure 3 cancers-13-03973-f003:**
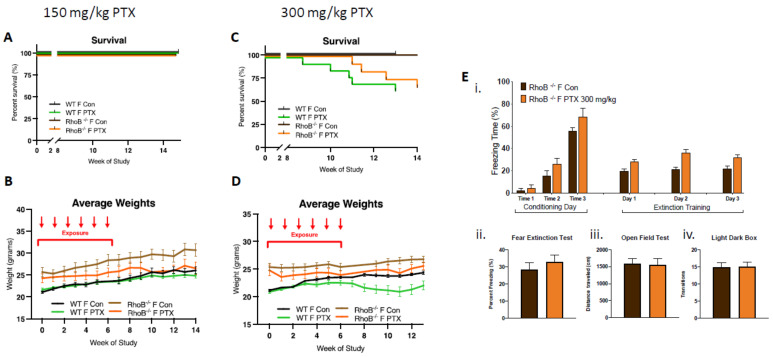
*rhoB* deficiency does not impact systemic toxicity or neurotoxicity in females treated with paclitaxel. (**A**) After 150 mg/kg paclitaxel treatment, survival curves showed no mortality in either WT (*n* = 18) or *rhoB-*deficient females (*n* = 12) as compared with control mice (WT + *rhoB*^–/–^ *n* = 21). (**B**) *rhoB-*deficient females showed no weight uptake as compared with control untreated *rhoB-*deficient females, whereas WT treated and untreated mice showed similar weight uptake. Three-way ANOVA of weights found significant variation between female *rhoB*^–/–^ and WT genotypes (*p* < 0.0001), paclitaxel treatments (*p* < 0.0001), and genotypes × treatments (*p* < 0.0001) over 14 weeks. Chart uses mean ± s.e.m. (**C**) After 300 mg/kg paclitaxel treatment, survival curves show 36% mortality in WT (*n* = 14) and 25% mortality in *rhoB-*deficient females (*n* = 12) 13 weeks post-paclitaxel. Comparison survival curves demonstrate significant difference (*p* < 0.0001) in log-rank (Mantel–Cox) test. (**D**) WT females showed less weight uptake from treatment. Three-way ANOVA of weights found significant variation due to treatment (*p* < 0.0001) and genotype (*p* < 0.0001). (**E**) 300 mg/kg paclitaxel administration, (**i**) acquisition of conditioned fear (three tone-shock pairings) was not impaired in *rhoB-*deficient female and mice did not exhibit decreased freezing behavior over extinction sessions (tone only) (*n* = 8). (**ii**–**iv**) No significant difference in fear extinction (*n* = 8), mobility (control *n* = 12, treated *n* = 12), or anxiety (control *n* = 12, treated *n* = 12) behavior in *rhoB-*deficient females after 300 mg/kg paclitaxel was found (unpaired *t*-test).

**Figure 4 cancers-13-03973-f004:**
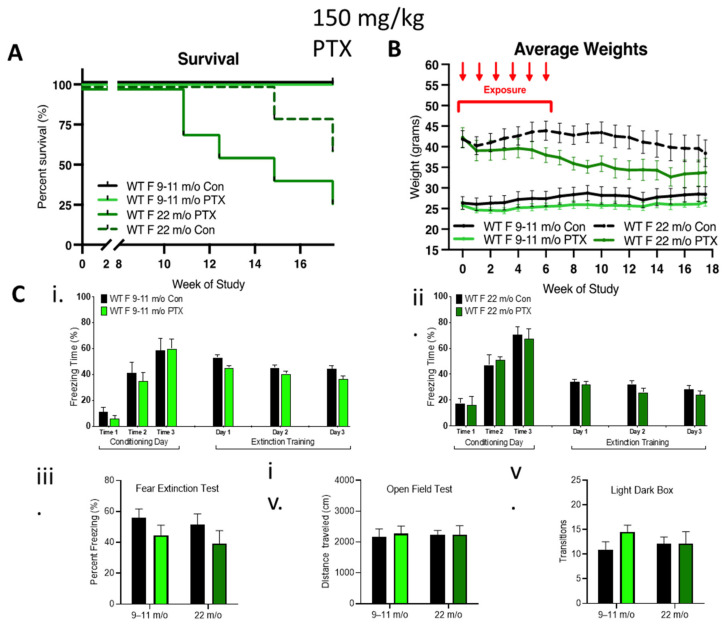
Effect of age on female tolerance to paclitaxel and protection from neurotoxicity induced by paclitaxel. (**A**) WT aged females were treated only with 150 mg/kg paclitaxel. The 9–11-month-old female mice (*n* = 7) were tolerant to paclitaxel, whereas 75% of older mice (22-month-old, *n* = 7) died over 18 weeks. Significant change in survival curve log-rank (Mantel–Cox) test (*p* = 0.021) was found. (**B**) The 9–11-month-old females showed stable weight as compared to controls, whereas very old females showed an age-related drop in weight that was enhanced in the paclitaxel group. Three-way ANOVA demonstrated a significant variation in age (*p* < 0.0001), treatment (*p* < 0.0001), and age x treatment (*p* = 0.0002). (**C**) Paclitaxel administration in old (9–11-month-old) WT female mice did not impair (**i**) the acquisition of conditioned fear (three tone-shock pairings). All mice showed a graduate decrease in freezing behavior over extinction sessions (tone only) (*n* = 7). (**ii**) Paclitaxel administration in very old WT female mice did not impair the acquisition of conditioned fear (three tone-shock pairings). All mice showed a graduate decrease in freezing behavior over extinction sessions (tone only) (control *n* = 5, treated *n* = 4). (**iii–v**) No significant difference (two-way ANOVA) in fear extinction (9–11-month-old control and treated *n* = 7, older control *n* = 5, treated *n* = 4), mobility (very old control *n* = 5, treated *n* = 6), or anxiety behavior (very old *n* = 5) in old or very old females after 150 mg/kg paclitaxel treatment. Graphs designate mean ± s.e.m.

**Figure 5 cancers-13-03973-f005:**
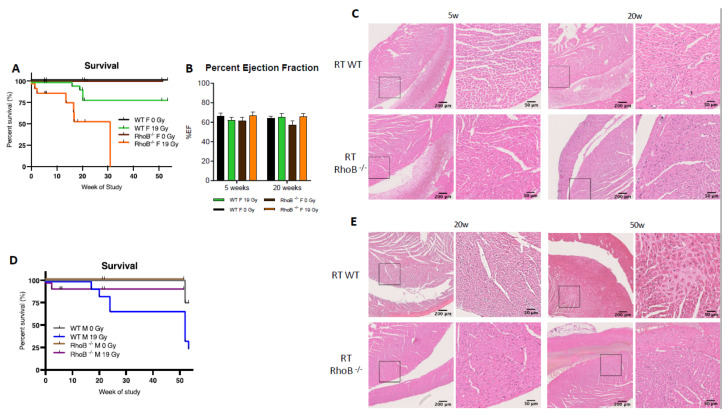
*rhoB* deficiency reverses cardioprotection in females exposed to RT. (**A**) *RhoB-*deficient females reached LD100 after 19 Gy thoracic at 30 weeks post-irradiation, whereas WT females exhibit 80% survival. Significant change between *rhoB*^–/–^ 0 Gy (*n* = 14), *rhoB^–/–^* 19 Gy (*n* = 18), WT 0 Gy (*n* = 32), and WT 19 Gy (*n* = 32), and female mice exposure survival curves (*p* < 0.0001) in log-rank (Mantel–Cox) test. (**B**) No change in %EF was observed in WT and *rhoB^–/–^* animals 5 and 20 weeks post-RT. (**C**) H&E staining analysis showed no alteration of the cardiac structure 5 weeks post-RT and enhanced leukocyte infiltration at 20 weeks. (**D**) *RhoB-*deficient males (*n* = 15) were protected from radiation-induced toxicity as compared to WT males (*n* = 9). (**E**) H&E staining analysis showed no histologic alteration or fibrosis up to 50 weeks post-RT in *rhoB*^–/–^ males wheras WT males showed significant alterations.

**Table 1 cancers-13-03973-t001:** Behavior Testing Timeline. WT = Wild Type animal, *rhoB^–/–^* = *rhoB* deficient animals, F = female, M = male, PTX = paclitaxel, OFT = Open Field Testing, LDB = Light Dark Box, and FX = Fear Extinction.

Paclitaxel Group			Age at Start of Study			Time of Behavioral Task During Study								
Genotype	Sex	Dose (mg/kg)	Age (Weeks)	*n* *Con*	*n* *PTX*	OFT (Week)	*n* *Con*	*n* *PTX*	LDB (Week)	*n* *Con*	*n* *PTX*	FX (Week)	*n* *Con*	*n* *PTX*
WT	F	150	**15**	11	18	**10**	11	18	**12**	7	8	**14**	8	8
	M		**14**	12	18	**11**	12	16	**13**	12	16	**14**	8	8
	F	300	**11**	14	14	**9**	14	13	**11**	14	11	**13**	8	8
	M		**13**	12	12	**-**	-	-	**-**	-	-	**8**	6	6
	F	150	**28**	7	7	**10**	7	7	**12**	7	7	**13**	7	7
			**81**	5	7	**10**	5	6	**12**	5	5	**13**	5	3
*rhoB^–/–^*	F	300	**25**	12	12	**17**	12	8	**17**	12	8	**18**	8	8

## Data Availability

Data will be made available on Zenodo, accessible on 1 November 2021.

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
