# Peer review of "Sex-Specific Differences in Toxicity Following Systemic Paclitaxel Treatment and Localized Cardiac Radiotherapy"

_cancers, 2021, doi:10.3390/cancers13163973_

Round 1
Reviewer 1 Report
This is an interesting and detailed study of the sensitisation effect of paclitaxel on radiation cardiomyopathy, with behavioural, functional and histological endpoints.
Abstract: the sentence beginning with "RhoB deficiency....." needs to be re-written in order to be understood.
Main text
In 2.3, a radiation dose is not given. 19 Gy is mentioned in some graphics. Is this is a single dose, or fractionated? Please state the expected effect of this when used alone.
In 2.6 begin this section with "Echocardiography was performed....."
Give formula using an equation editor or equivalent as a fromal equation.
In 2.9, normal distributions are assumed without confirmation; the alternative of skewed or nonparamteric distribution free tests should also be considered. Please justify your choice of statistical tests with cogent reasons.
Figure4 The caption should start "Effect of age on female tolerance...."
In some places please usual sex status to avoid potetnial confusion.
There appears to be no study of the male testosterone related biology. Is this also intended?
The authors may wish to speculate about the effects of oestrogen blockading drugs, or at least warn about them.
The other issue is the very limited life span of a mouse in terms of manifestation of so called 'late radiation effects', which may take years to develop in the human; is there any possibility that a different time lag to the manifestation of cardiac effects is related to the sex status of the animal? This possibility should be mentioned.
Also the modification of late effects by using fractionated radiation needs to be mentioned.
Author Response
We would like to thank the reviewers for their comments and have provided a point-by-point response to their critiques as detailed below. The modifications included in the MS have been highlighted in yellow for easy identification.
# expert 1 :
This is an interesting and detailed study of the sensitisation effect of paclitaxel on radiation cardiomyopathy, with behavioural, functional and histological endpoints.
Abstract: the sentence beginning with "RhoB deficiency....." needs to be re-written in order to be understood.
The sentence was modified accordingly:
In females, RhoB deficiency and advanced age had no impact on neurocognitive impairment induced by Paclitaxel but enhanced cardiac sensitivity to radiotherapy.
Main text
In 2.3, a radiation dose is not given. 19 Gy is mentioned in some graphics. Is this is a single dose, or fractionated? Please state the expected effect of this when used alone.
A single dose of 19Gy was used. The physiopathological impact of a single dose of 19Gy is shown in Figure 2 and mimics physiopathological features of late toxicities observed in human patients.
In 2.6 begin this section with "Echocardiography was performed....."
Give formula using an equation editor or equivalent as a fromal equation.
The formula for EF (%) = [(LVDA LVSA)/LVDA] X100 and has been added p4.
In 2.9, normal distributions are assumed without confirmation; the alternative of skewed or nonparamteric distribution free tests should also be considered. Please justify your choice of statistical tests with cogent reasons.
Normal distribution was used as our results are characterized by the mean and SE/SEM values, while the outcomes of the various assays depend upon numerous and independent parameters. In addition, the number of animals included was based on the necessity of ensuring sufficient power to detect group differences. Statistical analyses were chosen after consultation with the Biostatistician and adapted relatively to the assay performed. Therefore, as explained in the material and methods “unpaired, two-tailed tests were used to compare behavioral results with two groups. Two-way analysis of variance (ANOVA) was used to assess for interaction and group effects of more than two groups for behavioral and echocardiography testing, if significance was found, Bonferroni’s multiple comparisons test was conducted. Three-way ANOVA was used to compare four groups and two-way ANOVA was used to compare two groups of weight data, no multiple comparisons assessment was performed on weight data. Log-rank (Mantel-Cox) test was used to compare survival curves. P ≤ 0.05 was considered statistically significant.”
Figure4 The caption should start "Effect of age on female tolerance...."
Figure 4 caption has been modified according to the reviewer comment.
In some places please usual sex status to avoid potetnial confusion.
In the Figures, a color code was used with females in green and males in blue. Results obtained with Tg mice are shown in orange. In the results, subtitles clearly state when both males and females are used or females only, paragraph 3.3 for instance.
There appears to be no study of the male testosterone related biology. Is this also intended?
Yes this was the intent, as published studies have been mostly performed in male animals, see discussion P12 and reference 35.
The authors may wish to speculate about the effects of oestrogen blockading drugs, or at least warn about them.
The role of estrogen is clearly complex as stated in the discussion p12 and 13, additional specific studies using anti-estrogen drugs are needed to make more conclusive statements and are beyond the scope of the current manuscript.
The other issue is the very limited life span of a mouse in terms of manifestation of so called 'late radiation effects', which may take years to develop in the human; is there any possibility that a different time lag to the manifestation of cardiac effects is related to the sex status of the animal? This possibility should be mentioned.
This comment can apply to any experimental study with mice, however the irradiation regimen used (19 Gy single dose) was chosen to approximate the physiopathology of late toxicity observed in humans in a time-frame consistent with a mouse life-span. Note: than the mouse life-span is +/- 3 years. Our study was conducted over a year which represents 1/3 of a mouse life-span. This time frame is reasonably consistent with the development of late radiation effects in human patients.
Also the modification of late effects by using fractionated radiation needs to be mentioned.
As stated above, we aimed at approximating human physiopathology using a mouse model. To achieve this aim in mice, a single high dose of irradiation is the most efficient means to trigger cardiac toxicity in a time frame consistent with an experimental study. Note that implementing a fractionation regimen can frequently confound the interpretation results performed in mice as repeated anesthesia can interfere with outcome severity and onset, which was the rationale for avoiding this strategy in the current paper.
Reviewer 2 Report
The authors present a very interesting and well-written article.
I have only very few comments:
- Other tumor entities treated with paclitaxel (e.g. H&N cancer, esophageal cancer) should be mentioned.
- The authors should add a limitations paragraph to the discussion.
- The authors should describe in a more detailed way the potential benefits of their study for cancer patients and present their "vision" how their results could find the way into clinical routine.
Author Response
Point by point answer to the reviewers.
We would like to thank the reviewers for their comments and have provided a point-by-point response to their critiques as detailed below. The modifications included in the MS have been highlighted in yellow for easy identification.
# expert 2 :
The authors present a very interesting and well-written article.
I have only very few comments:
- Other tumor entities treated with paclitaxel (e.g. H&N cancer, esophageal cancer) should be mentioned.
According to the reviewer’s comment H&N and esophageal cancer were added in the introduction P2. In these indications, carboplatin is usually associated with paclitaxel + RT.
- The authors should add a limitations paragraph to the discussion.
This has now been included – see page 13.
“Further work is clearly required to unravel the mechanistic basis of our findings and explore how they might impact personalized treatments in clinical practice. Neurocognitive assessments were undertaken to survey the impact of systemic Paclitaxel across multiple brain regions, but the precise levels of Paclitaxel crossing the blood brain barrier into discrete regions and among the different age groups were not assessed. More importantly the impact of combined treatment (paclitaxel + radiotherapy) was not investigated whereas the combination is expected to enhance cardiac toxicity and possibly influence neurocognitive outcome by the release of paracrine factors. Future studies undertaking direct measurement of circulating and tissue specific estrogen levels along with other non-genetic targeted interventions aimed at dissecting fibrogenic signaling cascades would also provide deeper mechanistic insight into tissue remodeling processes affecting the heart and surrounding organs (esophagus, lung).”
- The authors should describe in a more detailed way the potential benefits of their study for cancer patients and present their "vision" how their results could find the way into clinical routine.
The first paragraph of the discussion addresses this question (see below). However, as our study is the first of a kind, we wanted to avoid too much speculation on the potential ramifications of our study, as additional investigations re clearly warranted.
“Surprisingly and despite the emergence of tailored treatments and personalized medicine, the impact of sex on therapeutic outcome and tolerance to treatment to date, have been poorly investigated. These investigations are of utmost relevance since sex specific differences may not only affect tumor biology but also the pharmacokinetics and dynamics of drugs and response to local treatment such as radiotherapy. In this context, the present experimental study was designed to investigate the impact of sex following two classical anti-cancer treatments i.e. paclitaxel and RT. Both are widely used and administered according to the type and grade of cancer, but importantly, not according to the sex of the patient.”